# Exploiting Broad-Spectrum Chimeric Lysin to Cooperate with Mupirocin against *Staphylococcus aureus*-Induced Skin Infections and Delay the Development of Mupirocin Resistance

Xiao-chao Duan,[a,b,c] Xin-xin Li,[a,b,c] Xiang-min Li,[a,b,c] Shuang Wang,[a,b,c] Fen-qiang Zhang,[a,b,c] Ping Qian[a,b,c]

aState Key Laboratory of Agricultural Microbiology, Huazhong Agricultural University, Wuhan, China
bCollege of Veterinary Medicine, Huazhong Agricultural University, Wuhan, China
cKey Laboratory of Preventive Veterinary Medicine in Hubei Province, The Cooperative Innovation Center for Sustainable Pig Production, Wuhan, China

Xiao-chao Duan and Xin-xin Li contributed equally to this work. Author order was determined on the basis of seniority.

**ABSTRACT** *Staphylococcus aureus* often leads to severe skin infections. However, *S. aureus* is facing a crisis of antibiotic resistance. The combination of phage and antibiotics is effective for drug-resistant *S. aureus* infections. Therefore, it is worth exploiting novel antibacterial agents to cooperate with antibiotics against *S. aureus* infections. Herein, a novel chimeric lysin ClyQ was constructed, which was composed of a cysteine- and histidine-dependent amidohydrolase/peptidase (CHAP) catalytic domain from *S. aureus* phage lysin LysGH15 and cell wall-binding domain (CBD) from *Enterococcus faecalis* phage lysin PlyV12. ClyQ had an exceptionally broad host range targeting streptococci, staphylococci, *E. faecalis*, and *E. rhusiopathiae*. ClyQ combined with mupirocin (2.64 log reduction) was more effective at treating *S. aureus* skin infections than ClyQ (0.46 log reduction) and mupirocin (2.23 log reduction) alone. Of equal importance, none of *S. aureus* ATCC 29213 or S3 exposed to ClyQ developed resistance, and the combination of ClyQ and mupirocin delayed the development of mupirocin resistance. Collectively, chimeric lysin ClyQ enriches the reservoirs for treating *S. aureus* infections. Our findings may provide a way to alleviate the current antibiotic resistance crisis.

**IMPORTANCE** *Staphylococcus aureus*, as an *Enterococcus faecium*, *Staphylococcus aureus*, *Klebsiella pneumoniae*, *Acinetobacter baumannii*, *Pseudomonas aeruginosa*, and *Enterobacter* species (ESKAPE) pathogen, can escape the elimination of existing antibiotics. At present, phages and phage lysins against *S. aureus* infections are considered alternative antibacterial agents. However, the development of broad-spectrum chimeric phage lysins to cooperate with antibiotics against *S. aureus* infections remains at its initial stage. In this study, we found that the broad-host-range chimeric lysin ClyQ can synergize with mupirocin to treat *S. aureus* skin infections. Furthermore, the development of *S. aureus* resistance to mupirocin is delayed by the combination of ClyQ and mupirocin *in vitro*. Our results bring research attention toward the development of chimeric lysin that cooperates with antibiotics to overcome bacterial infections.

**KEYWORDS** chimeric phage lysin, mupirocin, *S. aureus* infections, antibiotic resistance

Staphylococcus aureus, as the most common commensal bacteria on the skin (1), plays a major part in atopic dermatitis (AD), and more than 90% of AD cases are caused by *S. aureus* (2, 3). Most studies on *S. aureus* have been emphasized with the attention being given to the antimicrobial resistance of *S. aureus* (4). Furthermore, biofilms protect *S. aureus* from the clearance of antibiotics and the immune system (5).

Currently, phage lysins, such as *Staphylococcus* phage endolysin SAL-200 (6), antistaphylococcal lysin LSVT-1701 (7), and *Streptococcus suis* phage lysin CF-301 (8), have

Address correspondence to Ping Qian, qianp@mail.hzau.edu.cn.

The authors declare no conflict of interest.

been successfully applied to treating clinical bacterial infections. As expected, this has boosted the study of phage lysins and chimeric lysins as new antimicrobial agents in *S. aureus*-caused infections (9, 10). Notably, previous studies demonstrated that CF-301 has synergistic antibacterial activity against *S. aureus* with antibiotics *in vitro* and *in vivo* (11). The combination of endolysin MR-10 and minocycline could treat methicillin-resistant *S. aureus* (MRSA)-induced wound infection (10). Moreover, both phage lysin PlySs2 and chimeric lysin ClyS did not cause the development of *S. aureus* and *S. pneumoniae* resistance within 8 days (12, 13). Additionally, the combination of CF-301 and daptomycin delayed the development of *S. aureus* MW2 resistance to daptomycin after 26 days of passage (14). Although numerous studies have demonstrated that chimeric lysins possess perfect activity against *S. aureus in vivo* and *in vitro* (15–17), there are no reports on a chimeric lysin that yields a surprising broad-spectrum activity and exhibits a synergistic antibacterial effect with antibiotics.

In this study, we constructed a novel chimeric lysin ClyQ, which possesses antibacterial activity against staphylococci, streptococci, *Enterococcus faecalis*, and *Erysipelothrix rhusiopathiae in vitro*. Our results revealed that ClyQ could significantly remove the biofilms of *S. aureus*. Meanwhile, ClyQ contributed to treating *S. aureus* systemic and skin infections. Remarkably, the combination of ClyQ and mupirocin was superior to ClyQ and mupirocin treatment alone for removing *S. aureus* from the skin and delaying the development of resistance to mupirocin.

## RESULTS

**Construction, expression, and purification of ClyQ.** To solve the limited host range of phage lysins, we constructed a novel chimeric lysin ClyQ. ClyQ was composed of a cysteine- and histidine-dependent amidohydrolase/peptidase (CHAP) domain from *S. aureus* phage lysin LysGH15 and a cell wall-binding domain (CBD) from *E. faecalis* phage lysin PlyV12, and the two domains were ligated by a glycine-serine (GGSSGS) linker (Fig. S1A in the supplemental material). The molecular weight of purified ClyQ was approximately 38.0 kDa (Fig. S1B).

**Stability of ClyQ.** Our results showed that the lytic activity of ClyQ could not be significantly changed at 37 to 42.5°C. When ClyQ was incubated at 45°C, the activity of ClyQ began to decrease and was completely lost at 60 min (Fig. S2A), and the protein melting temperature ($T_m$) of ClyQ was 43.9°C (Fig. S2B). Concentrations of 10, 100, and 1,000 $\mu$M $Ca^{2+}$, $Mn^{2+}$, $Mg^{2+}$, and $Ba^{2+}$ could promote the lytic activity of ClyQ. Simultaneously, the strength of 10 $\mu$M $Zn^{2+}$ could not affect ClyQ, although 100 and 1,000 $\mu$M $Zn^{2+}$ inhibited the lytic activity of ClyQ (Fig. S2C). Furthermore, ClyQ could maintain complete activity at pH 7 to 10. The activity of ClyQ was reduced by approximately 23.3% after incubation in phosphate-buffered saline (PBS; pH 11) for 60 min, and ClyQ had little activity at pH 5 to 6 (Fig. S2D).

**ClyQ exhibits excellent antibacterial effects *in vitro*.** LysGH15, as a *S. aureus* phage lysin, possesses remarkable lytic activity against *S. aureus* (18). ClyQ shares the same CHAP domain as LysGH15. Thus, the host range and antibacterial activity of ClyQ were compared with those of LysGH15. As shown in Fig. 1A, 9 strains of *S. aureus* were sensitive to ClyQ and LysGH15, and the turbidity reduction ratio of ClyQ ranged between 50.29 and 88.51% for these *S. aureus* strains. For other staphylococci (including *S. simulans*, *S. cohnii*, *S. epidermidis*, *S. heamolyticus*, *S. warneri*, *S. chromogenes*, *S. xylosus*, *S. muscae*, *S. nepalensis*, *S. equorum*, *S. rostri*, *S. saprophyticus*, *S. arlettae*, and *S. hyicus*), the turbidity reduction ratio of ClyQ was in the range of 18.84 to 90.72% and that of LysGH15 was in the range of 14.96 to 88.23%. For streptococci (including *S. agalactiae*, *S. dysgalactiae*, *S. uberis*, and *S. suis*), the turbidity of these strains was reduced by 21.89 to 64.79% (Fig. 1A). For *E. faecalis* and *E. rhusiopathiae*, the optical density at 600 nm ($OD_{600}$) values could be decreased by ClyQ, but *Listeria monocytogenes*, *Escherichia coli*, and *Salmonella enterica* serovar Typhimurium were not sensitive to ClyQ (Fig. 1A). Additionally, LysGH15 could not lyse the streptococci *E. faecalis*, *E. rhusiopathiae*, *Listeria monocytogenes*, *Escherichia coli*, and *Salmonella* Typhimurium strains.

At the same time, we evaluated the antibacterial activity of ClyQ and LysGH15 against staphylococci *in vitro* because staphylococci are the primary opportunistic pathogens in

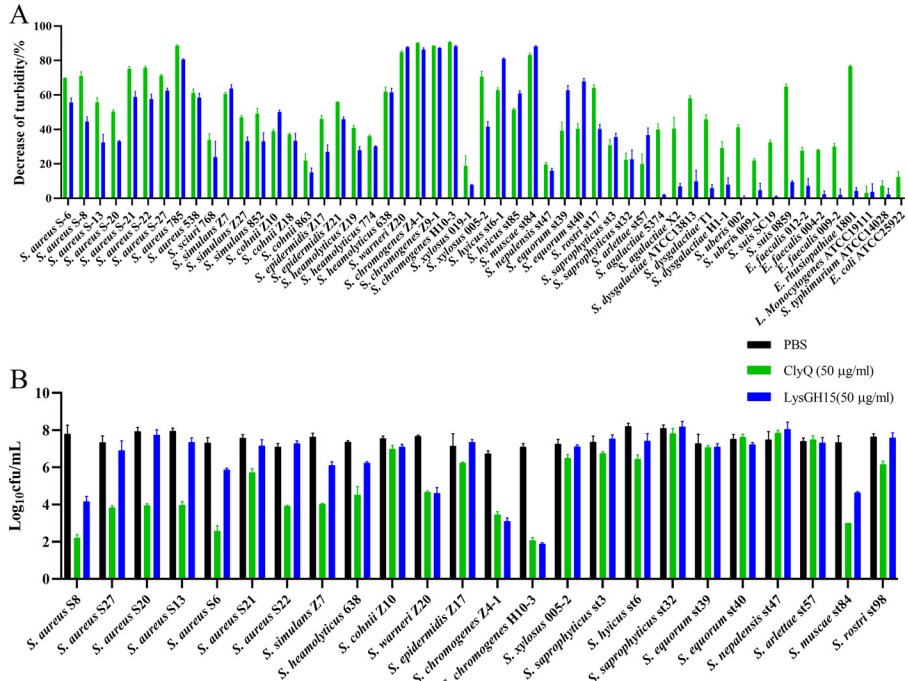

**FIG 1** The host range and antistaphylococcal activity of chimeric lysin ClyQ and *S. aureus* phage lysin LysGH15 *in vitro*. (A) Host range of ClyQ and LysGH15. Different bacterial species (including staphylococci, streptococci, *E. faecalis*, *E. rhusiopathiae*, *L. monocytogenes*, *E. coli*, and *S.* Typhimurium) were used to determine the host range of ClyQ and LysGH15. (B) The antistaphylococcal activity of ClyQ and LysGH15. Different species of *Staphylococcus* (including *S. aureus*, *S. simulans*, *S. cohnii*, *S. epidermidis*, *S. heamolyticus*, *S. warneri*, *S. chromogenes*, *S. xylosus*, *S. muscae*, *S. nepalensis*, *S. equorum*, *S. rostri*, *S. saprophyticus*, *S. arlettae*, and *S. hyicus*) were used to test the antistaphylococcal activity of ClyQ and LysGH15. All experiments were repeated three times. The error bars show the standard deviations (SD).

the skin (19). Specifically, ClyQ and LysGH15 could reduce the visible bacteria number of *S. aureus* in bacteria suspensions by 1.86 to 5.59 log and 0.00 to 3.63 log, respectively (Fig. 1B). Moreover, other staphylococci strains were reduced by 0.00 to 5.04 log and 0.00 to 5.21 log, respectively (Fig. 1B). Our results demonstrate that ClyQ possesses a broad host range and exceptional lytic activity.

We examined the lytic activity of ClyQ under various times and concentrations. Our data indicated that ClyQ could lyse *S. aureus* ATCC 43300 and S3 in a dose-dependent manner (Fig. 2A and B). At the same time, ClyQ (50 μg/mL) could continuously reduce the visible number of ATCC 43300 and S3 within 60 min (Fig. 2C and D). Specifically, after 60 min of incubation, the bacterial numbers of ATCC 43300 and S3 in the ClyQ-treated bacteria suspensions were reduced by 2.46 and 4.12 log, respectively (Fig. 2C and D).

Furthermore, the MIC of ClyQ against *S. aureus* strains ranged from 4 to 16 μg/mL and that of *S. equorum*, *S. hyicus*, *S. muscae*, *S. rostri*, *S. sciuri*, *S. simulans*, *S. cohnii*, *S. simulans*, and *S. warneri* ranged from 2 to 16 μg/mL. Meanwhile, the MIC for the *S. arlettae*, *S. nepalensis*, *S. saprophyticus*, and *S. heamolyticus* strains was greater than 128 μg/mL (Table 1).

**ClyQ effectively clears the biofilms of *S. aureus*.** *S. aureus* can colonize and form biofilms on skin infection sites, contributing to antibiotic resistance (20, 21). As a result, crystal violet (CV) staining and colony count methods were used to evaluate the effect of ClyQ on clearing *S. aureus*-formed biofilms. On the one hand, the results of CV staining indicated that 24-h and 48-h biofilms of *S. aureus* ATCC 43300 were cleared after ClyQ (40 μg/mL) treatment by 62.27 and 61.28%, respectively (Fig. 3A), and 40 μg/mL ClyQ could clear 24-h and 48-h biofilms of *S. aureus* S3 by 58.01 and 71.04%, respectively (Fig. 3B). On the other hand, as shown in Fig. 3C, ClyQ at a concentration of 40 μg/mL could decrease the number of sessile *S. aureus* ATCC 43300 in biofilms by

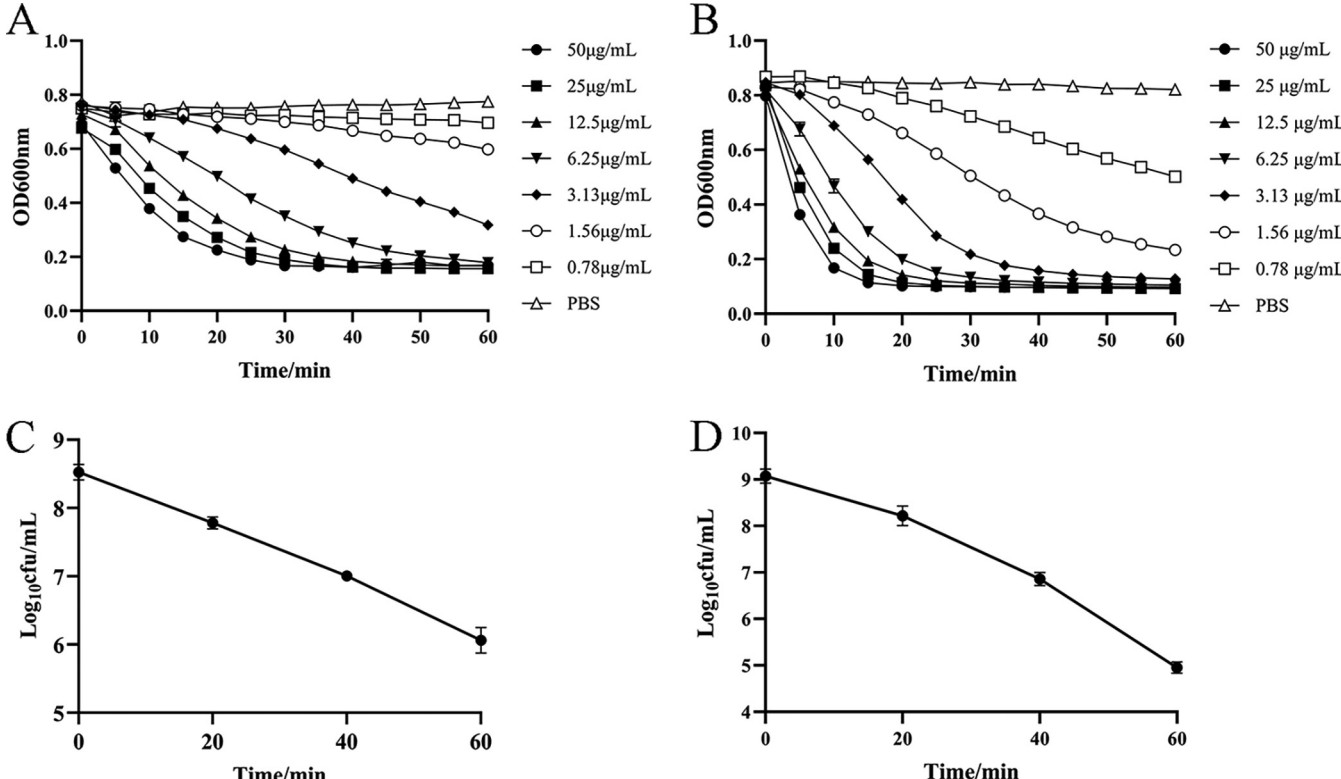

**FIG 2** Bactericidal activity of ClyQ against *S. aureus* ATCC 43300 and S3. (A and B) Lytic activity of ClyQ against *S. aureus* ATCC 43300 (A) and S3 (B) at different strengths (50, 25, 12.5, 6.25, 3.13, 1.56, 0.78, and 0 $\mu$g/mL) within 60 min. (C and D) Time-dependent bactericidal activity of ClyQ (50 $\mu$g/mL) against *S. aureus* ATCC 43300 (C) and S3 (D) within 60 min. All experiments were repeated three times. The error bars show the standard deviations (SD).

1.95 log (24-h biofilm) and 1.54 log (48-h biofilm). The number of sessile *S. aureus* S3 was decreased after treatment with 40 $\mu$g/mL ClyQ by 0.95 log (24-h biofilm) and 2.07 log (48-h biofilm) (Fig. 3D). In addition, field emission scanning electron microscope (FESEM) images showed that ClyQ could remove sessile bacteria and biomass in biofilms (Fig. 4). These data indicate that ClyQ can effectively clear *S. aureus*-formed biofilms in a concentration-dependent manner.

**ClyQ rescues mice from *S. aureus* systemic infection.** We identified that $8.0 \times 10^8$ CFU/mouse was the minimum lethal dose ($MLD_{100}$) for *S. aureus* ATCC 29213 (Fig. 5A). Subsequently, a mouse systemic infection model was established by intraperitoneal injection of $1 \times MLD_{100}$ ATCC 29213. At 1 h and 3 h postinfection, a vast amount of *S. aureus* ATCC 29213 was isolated from the heart (6.60 and 5.99 log), liver (7.41 and 7.50 log), spleen (8.23 and 8.15 log), lung (7.04 and 6.69 log), kidney (7.02 and 7.04 log), and blood (2.84 and 3.42 log) (Fig. 5B). Meanwhile, at 1 h postinfection, the survival rates of mice injected intraperitoneally with 200 $\mu$g/mouse ClyQ, 500 $\mu$g/mouse ClyQ, and 15 mg/kg vancomycin were 40%, 80%, and 100%, respectively (Fig. 5C). The survival rates of mice treated with 500 $\mu$g/mouse ClyQ, 1 mg/mouse ClyQ, and 15 mg/kg vancomycin at 3 h postinfection were 0%, 60%, and 40%, respectively (Fig. 5D).

**A combination of ClyQ and mupirocin exhibits a synergistic bactericidal effect on treating MRSA-induced skin infection.** *S. aureus*, especially MRSA, is a crucial pathogen causing skin infections (22, 23). The *S. aureus*-induced skin infection model was used in this study to evaluate the bactericidal efficacy of the combination of ClyQ and mupirocin. The results showed that ClyQ, mupirocin, and the combination of ClyQ and mupirocin could decrease the bacteria numbers on skin by 0.46 ($P < 0.05$), 2.23 ($P < 0.001$), and 2.64 ($P < 0.001$) log, respectively, compared to the PBS-treated group (Fig. 6A). A previous study demonstrated that *S. aureus* skin infection contributes to skin inflammation (24). Here, in the PBS-treated group, the concentrations of tumor necrosis factor-$\alpha$ (TNF-$\alpha$) and interleukin-6 (IL-6) at the site of *S. aureus* infection reached

**TABLE 1** MIC values of ClyQ

| Strains | Species of strains | MIC ($\mu$g/mL) |
|---|---|---|
| st57 | S. arlettae | >128 |
| st84 | S. muscae | 2 |
| st47 | S. nepalensis | >128 |
| st98 | S. rostri | 1 |
| st74 | S. saprophyticus | >128 |
| st60 | S. saprophyticus | >128 |
| Z4 | S. sciuri | 8 |
| Z7 | S. simulans | 2 |
| Z10 | S. cohnii | 8 |
| Z27 | S. simulans | 8 |
| Z20 | S. warneri | 16 |
| 638 | S. heamolyticus | >128 |
| ATCC 43300 | S. aureus | 16 |
| ATCC 29213 | S. aureus | 4 |
| S3 | S. aureus | 8 |
| st95 | S. aureus | 4 |
| st18 | S. aureus | 8 |
| st76 | S. aureus | 4 |
| st80 | S. aureus | 4 |
| st39 | S. equorum | 2 |
| st40 | S. equorum | 2 |
| st156 | S. equorum | >128 |
| st6 | S. hyicus | 2 |
| st85 | S. hyicus | 8 |
| st96 | S. hyicus | 8 |

430.69 and 395.79 pg/mL, respectively (Fig. 6B and C). For the control group, the concentrations of TNF-$\alpha$ and IL-6 were 56.8 and 92.9 pg/mL, respectively. Concentrations of TNF-$\alpha$ and IL-6 were significantly decreased by ClyQ treatment compared to in the PBS-treated group. Likewise, mupirocin and a combination of ClyQ and mupirocin could also reduce the levels of TNF-$\alpha$ and IL-6 (Fig. 6B and C).

A histopathological assay was used to evaluate the degree of tissue structure damage. The skin structure of the PBS-treated group was severely abnormal, the dermis was exposed, and the tissue was infiltrated with a high number of inflammatory cells (Fig. 6D). The tissue damage of mupirocin-treated and ClyQ-treated groups was alleviated compared to in the PBS-treated group (Fig. 6D). The combination of ClyQ and mupirocin had the best effect on skin infection. In detail, an abscess in the epidermis was observed, and inflammatory cell infiltration was not significant (Fig. 6D).

Our results showed that the combination of ClyQ and mupirocin had a synergistic antibacterial effect on treating skin infections (Fig. 6A). To determine whether ClyQ and mupirocin have a synergistic bactericidal effect, the fractional inhibitory concentration index (FICI) values of the combination of ClyQ and mupirocin were monitored. The combination of ClyQ and mupirocin against *S. aureus* S3, ATCC 43300, and ATCC 29213 yielded FICI values of 0.281 (Fig. 7A), 0.258 (Fig. 7B), and 0.531 (Fig. 7C), respectively.

**ClyQ combined with mupirocin delays the development of resistance to mupirocin.** In the study, we explored the development of resistance of *S. aureus* ATCC 29213 and S3 to ClyQ and mupirocin. After exposure of the 8th generation of ATCC 29213 to ClyQ, the MIC of ClyQ had not changed (Fig. 8A). After exposure of the 8th generation of S3 to ClyQ, the MIC of ClyQ increased 2-fold (Fig. 8B). In contrast, in the 4th, 5th, and 6th generations of ATCC 29213, the MIC of mupirocin increased surprisingly 2-, 128-, and 256-fold, respectively (Fig. 8A), and in the 3rd and 5th generations of S3, the MIC of mupirocin increased surprisingly 2- and 256-fold, respectively (Fig. 8B). When ClyQ was combined with mupirocin to induce the development of ATCC 29213 resistance *in vitro*, the MIC of ClyQ against ATCC 29213 had not changed in the 8th generation, and the MIC of mupirocin against ATCC 29213 only increased by 64-fold in the 7th generation of ATCC 29213 (Fig. 8C). Similarly, the MIC values of ClyQ in the 5th, 7th, and 8th generations of S3 increased by 2-, 4-, and 4-fold, respectively, and those for mupirocin

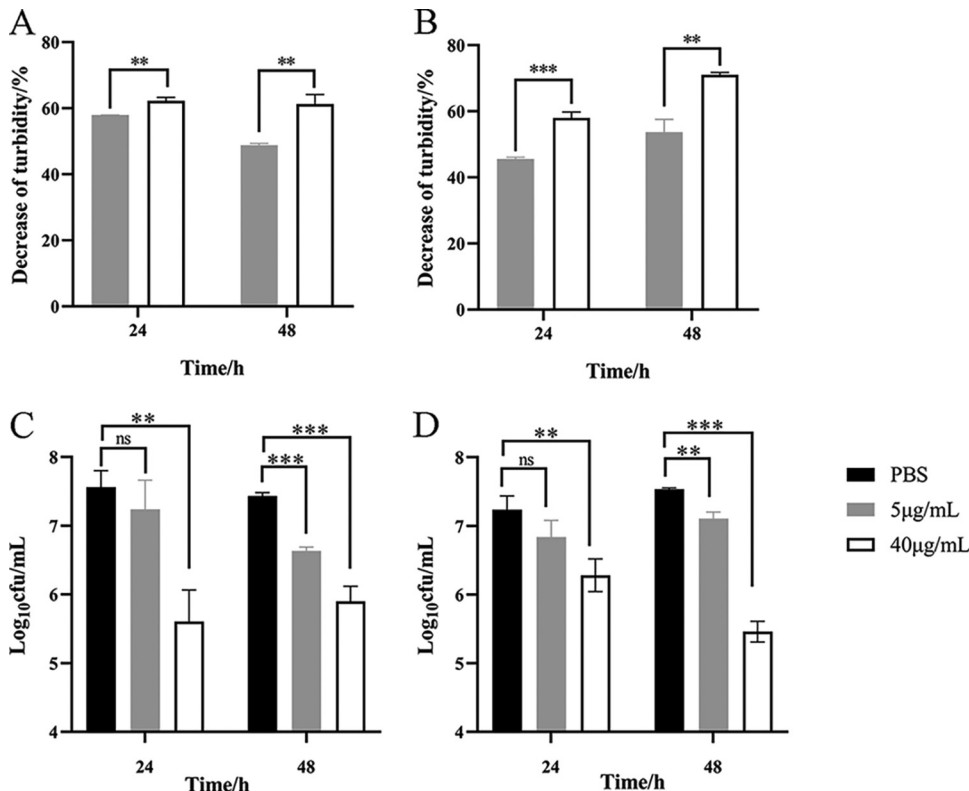

**FIG 3** The antibiofilm activity of ClyQ. (A and B) The biofilms of *S. aureus* ATCC 43300 (A) and S3 (B) were removed by treatment with ClyQ (5 and 40 μg/mL) at 37°C for 1 h. (C and D) Viable cell counts of *S. aureus* ATCC 43300 (C) and S3 (D) biofilms treated with ClyQ (5 and 40 μg/mL) at 37°C for 1 h. Significant differences between the PBS groups and the 5 μg/mL ClyQ or 40 μg/mL ClyQ group were determined by Student's *t* test; ns, not significant; **, $P < 0.01$; ***, $P < 0.001$. Error bars show the standard deviations (SD).

in the 3rd and 5th generations of S3 only increased by 8- and 16-fold, respectively (Fig. 8D).

## DISCUSSION

In this study, we designed a novel chimeric lysin ClyQ that possesses a broad host range (including staphylococci, streptococci, *E. faecalis*, and *E. rhusiopathiae*) and excellent antibacterial activity against staphylococci *in vitro*. Similarly, chimeric lysin ClyR can lyse *E. faecalis* and some species of staphylococci and streptococci (25). However, chimeric lysin ClyS exhibits exceptional muralytic activity against staphylococci (especially *S. aureus*) but little activity against streptococci (26). Chimeric lysin ClyV possesses lytic activity toward streptococci but not *S. aureus* and *E. faecalis* (27). In addition, ClyS is composed of a catalytic domain of *S. aureus* Twort phage lysin and a CBD from *S. aureus* phage lysin phiNM3 (26), and ClyR was constructed by fusing the catalytic domain of *Streptococcus* phage PlyC with a CBD from *Streptococcus* phage PlySs2 (25). Meanwhile, ClyV was constructed by fusing a catalytic domain of *Streptococcus* phage lysin PlyGBS with a CBD from *E. faecalis* phage lysin PlyV12 (27). In contrast, ClyQ was composed of a catalytic domain from *S. aureus* phage lysin LysGH15 and a CBD of *E. faecalis* phage lysin PlyV12.

Synergy between phage or phage lysins and antibiotics *in vitro* and *in vivo* has been reported (28). In our study, the combination of chimeric lysin ClyQ and mupirocin demonstrated superiority to ClyQ or mupirocin alone in removing *S. aureus* from the skin at 43 h postinfection (Fig. 6A). This agrees with other studies that showed that a combination of endolysin MR-10 (50 μg) and minocycline (50 mg/kg orally) is superior to single MR-10 (50 μg) or minocycline (50 mg/kg orally) treatment of MRSA-induced wound infection at 3 h postinfection (10). Combination therapy with *Streptococcus* phage lysin

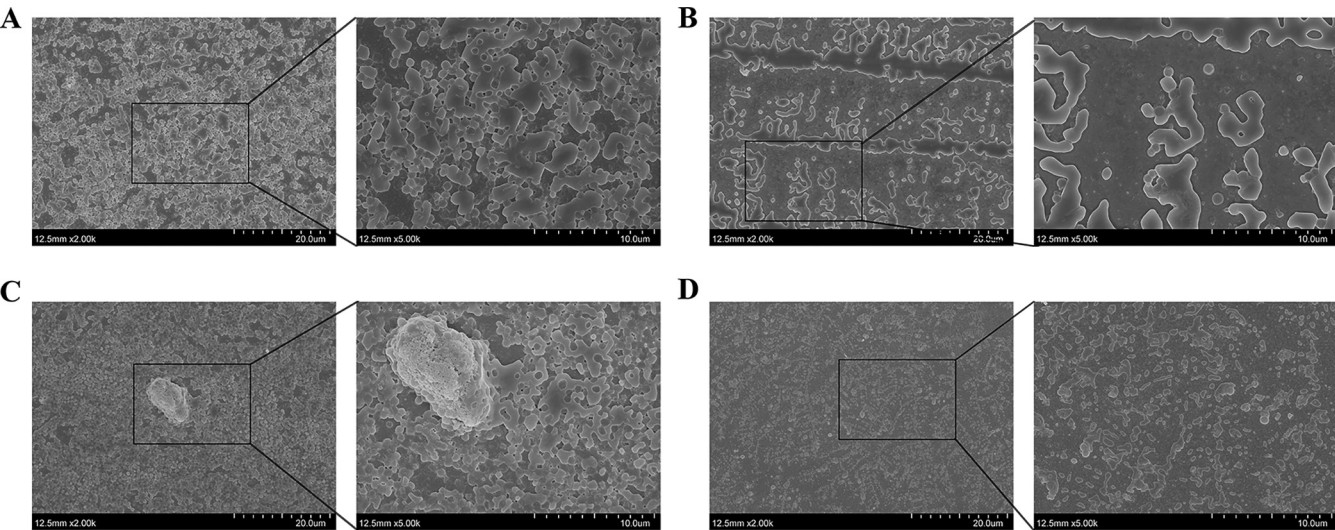

**FIG 4** Field emission scanning electron microscope images of biofilms formed by *S. aureus* ATCC 43300 and S3. The magnifications of the images are ×2,000 (left images) and ×5,000 (insets). The 48-h biofilms formed by *S. aureus* S3 were treated with PBS (A) and ClyQ (B), and the 48-h biofilms formed by *S. aureus* ATCC 43300 were treated with PBS (C) and ClyQ (D).

CF-301 and antibiotics (including daptomycin, vancomycin, and oxacillin) is superior to CF-301 and antibiotic monotherapy for treating *S. aureus*-induced mouse bacteremia (14). Furthermore, the combination of daptomycin (0.4 mg/kg) and pneumococcal phage lysin Cpl-1 (0.4 mg/kg) significantly increased the percentage of surviving mice infected with *S. pneumoniae* D39 compared with single daptomycin (0.4 mg/kg) or Cpl-1 (0.4 mg/kg) treatment (29). In contrast, the chimeric lysin ClyS (10% [wt/wt] in Aquaphor) is more effective than mupirocin (2% [wt/wt] in Aquaphor) in removing *S.*

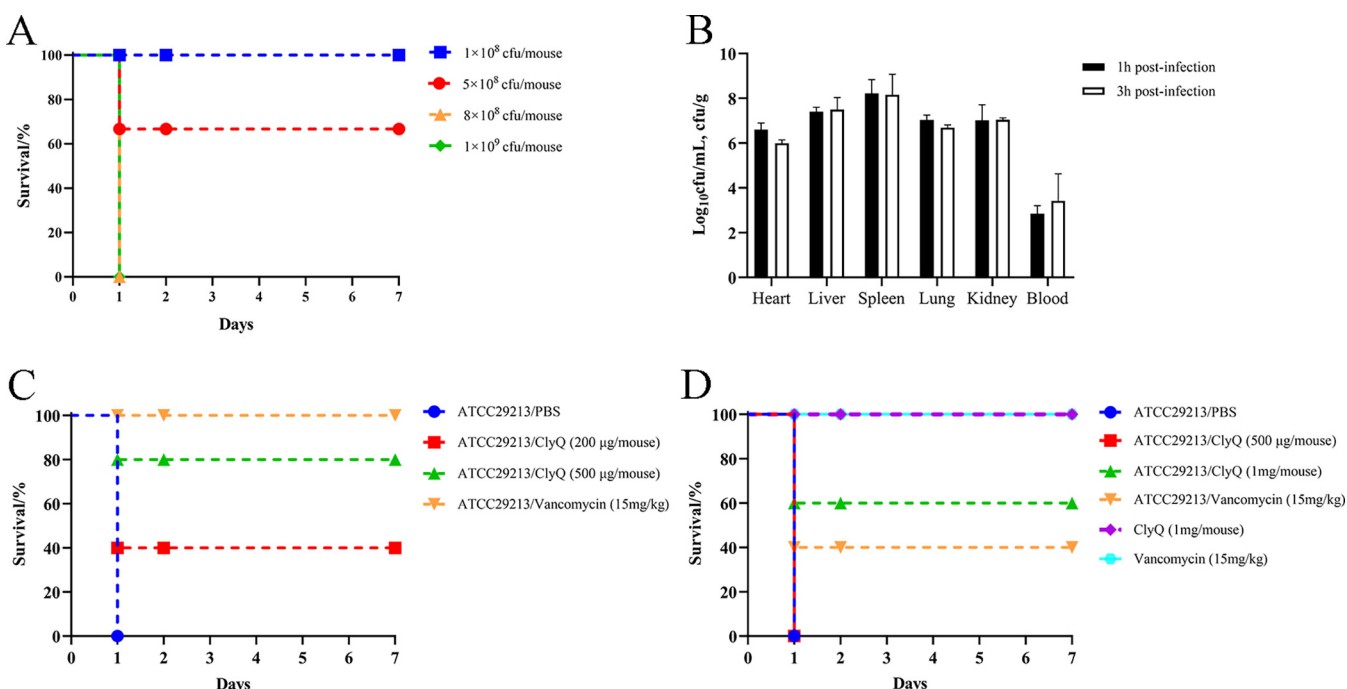

**FIG 5** Chimeric lysin ClyQ showed remarkable therapeutic efficiency in the *S. aureus* systemic infection model. (A) The $MLD_{100}$ of *S. aureus* ATCC 29213 in mice. The survival of mice infected with different concentrations of *S. aureus* ATCC 29213 was observed for 7 days. (B) Bacteria abundances in the heart, liver, spleen, lung, kidney, and blood of mice intraperitoneally infected with $1 \times MLD_{100}$ *S. aureus* ATCC 29213 at 1 h and 3 h postinfection. (C and D) Effect of ClyQ and vancomycin on the survival rate of mice infected with $1 \times MLD_{100}$ *S. aureus* ATCC 29213. Mice were intraperitoneally treated with PBS, vancomycin (15 mg/kg), and different concentrations of ClyQ at 1 h (C) and 3 h (D) postinfection. The survival rates of mice intraperitoneally injected with ClyQ and vancomycin alone were also recorded for 7 days. Data are expressed as means ± SD.

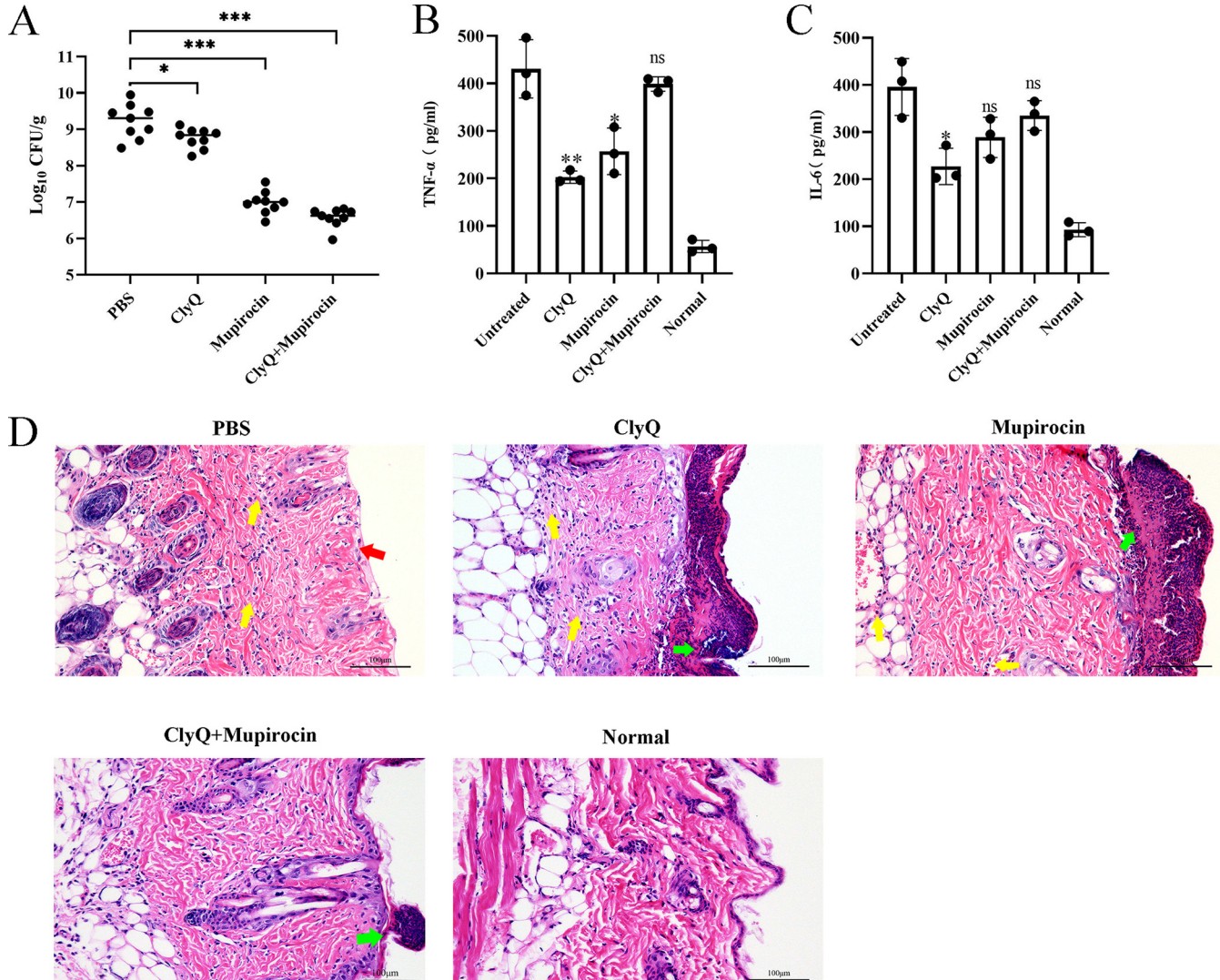

**FIG 6** The combination of ClyQ and mupirocin could effectively treat *S. aureus*-induced skin infection. The four groups of mice were treated with ClyQ, mupirocin, and a combination of ClyQ and mupirocin, respectively, at 20, 23, 26, 36, 39, and 42 h after infection. (A) The bacterial burden of skin tissue was determined at 43 h postinfection. (B and C) Levels of inflammatory cytokines TNF-$\alpha$ (B) and IL-6 (C) in the skin of mice were measured. (D) Hematoxylin and eosin staining of skin samples was observed by microscopic examination (magnification, $\times$200).

*aureus* colonized on the skin (12). However, ClyQ also exhibited potent synergy *in vitro*. In contrast to ClyQ and mupirocin monotherapy, the combination of ClyQ and mupirocin could delay the development of resistance to mupirocin (Fig. 8). Previous research has also reported that vancomycin and daptomycin resistance is suppressed by growth in combination with lysin CF-301 (14). To our knowledge, ClyQ is the first chimeric lysin to exhibit the ability to delay antibiotic resistance.

FICI values can determine whether there is a synergistic or adjuvant effect between antibacterial agents (30). Therefore, we measured the FICI values of the combination of chimeric lysin ClyQ and mupirocin. Our results showed that the FICI values of the combination of ClyQ and mupirocin against *S. aureus* ATCC 43300 and S3 were less than 0.5. However, the FICI values of the combination of chimeric lysin ClyQ and mupirocin against *S. aureus* ATCC 29213 was 0.531. Similarly, although the superiority of antistaphylococcal endolysin SAL-200 combinations with antibiotics was confirmed when treating *S. aureus* infection, the effect of the combination of SAL-200 and nafcillin on ATCC 29213 yielded an adjuvant effect (FICI of 0.533) by FICI test (6). In summary, the results indicate that a combination of ClyQ and mupirocin exhibits synergistic antibacterial activity against *S. aureus* skin infections.

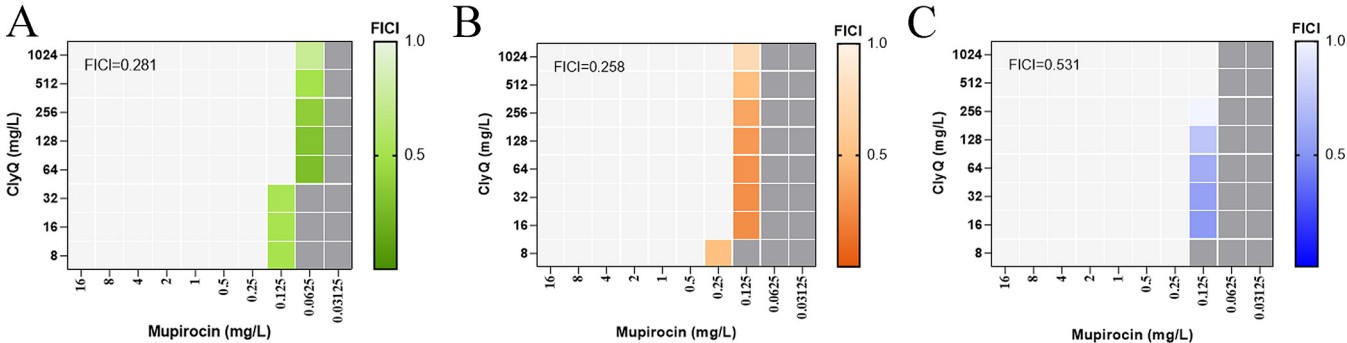

**FIG 7** Results of the checkerboard assay for the combination of ClyQ and mupirocin against *S. aureus*. (A to C) The synergistic effect of the combination of ClyQ and mupirocin against *S. aureus* S3 (A), ATCC 43300 (B), and ATCC 29213 (C) was tested. PBS was used as a negative control. The FICI values were calculated and are shown in the figures.

Surprisingly, for ClyQ, mupirocin, and a combination of ClyQ and mupirocin groups, the concentration of cytokines (TNF-$\alpha$ and IL-6) was greatly reduced, while the tendency of decreasing cytokine values was the opposite of the tendency of decreasing visible bacteria numbers (Fig. 6). The cytokine TNF-$\alpha$, as a proinflammatory cytokine, is produced soon after infection and promotes the acute inflammatory reaction. In the early stages of infection, TNF-$\alpha$ and IL-6 can activate neutrophils (31), and the more

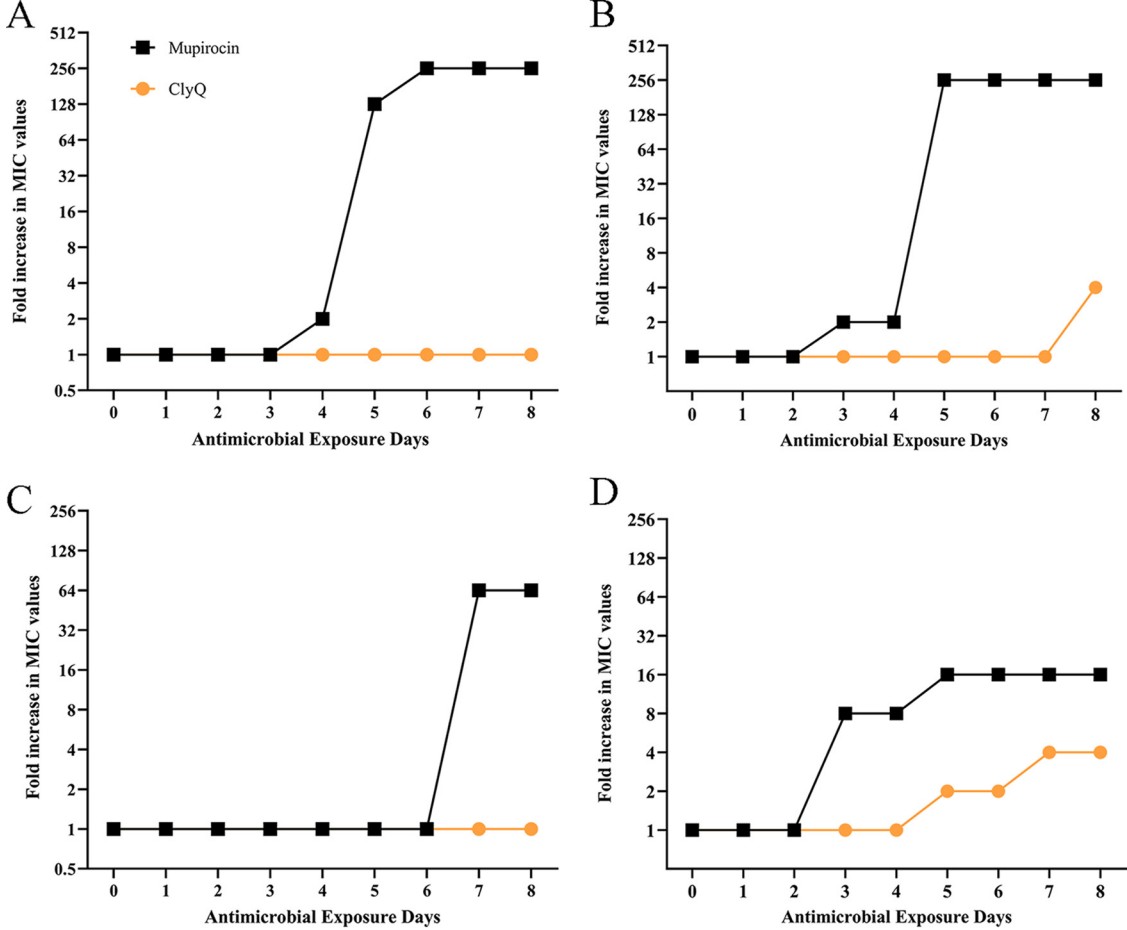

**FIG 8** Combination of ClyQ and mupirocin could delay the development of *S. aureus* resistance. (A and B) MIC changes of ClyQ and mupirocin against *S. aureus* ATCC 29213 (A) and S3 (B) when *S. aureus* was exposed to ClyQ or mupirocin alone. (C and D) MIC changes of ClyQ and mupirocin against *S. aureus* ATCC 29213 (C) and S3 (D) when *S. aureus* was exposed to the combination of ClyQ and mupirocin.

rapid lytic activity of lysins may lead to more peptidoglycan fragments, which provoke severe proinflammatory reactions (32). In our previous work, we have also discussed that Toll-like receptors (TLRs) were sensitive to *S. aureus* lipoproteins and nucleic acids, which could affect the production of proinflammatory cytokines (33). Based on these results, a possible reason could be that the combination of ClyQ and mupirocin could lyse more *S. aureus* cells than treatment with ClyQ and mupirocin alone, which causes more peptidoglycan fragments, lipoproteins, and nucleic acids to be released. Finally, these peptidoglycan fragments, lipoproteins, and nucleic acids could be recognized by TLRs and produce more TNF-$\alpha$ and IL-6.

Overall, a novel chimeric lysin ClyQ was constructed and showed exceptional anti-staphylococcal activity *in vitro* and *in vivo*. ClyQ also possesses the potential as an antimicrobial agent against biofilms formed by *S. aureus*. Importantly, the combination of ClyQ and mupirocin exhibits synergistic effects in a model of *S. aureus*-induced skin infections. The combination could delay the development of *S. aureus* resistance. In conclusion, our findings may enhance the development of new chimeric lysins as antibacterial agents that cooperate with antibiotics to combat *S. aureus*-induced skin infections and delay resistance development during treatment.

## MATERIALS AND METHODS

**Bacteria strains and culture conditions.** All bacteria strains in this study are listed in Table S1 in the supplemental material. *Staphylococcus* strains were grown and shaken in lysogeny broth (LB), and the other strains were grown in tryptic soy broth (TSB) containing 5% (vol/vol) bovine serum. *Escherichia coli* BL21(DE3) was used for protein expression and was grown in LB with kanamycin (50 $\mu$g/mL).

**Construction of the chimeric lysin ClyQ.** Genes of the full-length LysGH15 (GenBank: ADG26756.1) and PlyV12 lysins (GenBank: AAT01859.1) were synthesized by Tsingke Biotechnology, Co., Ltd. ClyQ was constructed by fusing the putative CHAP (from LysGH15; 1 to 165 amino acids) with CBD (from PlyV12; 244 to 309 amino acids). The ClyQ gene was digested with BamHI and HindIII and was subsequently ligated into the pET-28a(+) expression plasmid to construct pET-28a(+)-ClyQ. All primers are listed in Table S2.

**Expression and purification of ClyQ.** The plasmid pET-28a(+)-ClyQ was transferred into *E. coli* BL21 (DE3) cells and cultured in LB with 50 $\mu$g/mL kanamycin (37°C, 200 $\times$ *g*) to logarithmic phase (OD$_{600}$ of ~0.6). The recombinant plasmid was induced by the addition of 0.8 mM isopropyl-$\beta$-D-thiogalactopyranoside (IPTG) for 16 to 18 h at 16°C. The induced bacterial solution was centrifuged at 4°C for 10 min (6,000 $\times$ *g*) and resuspended with binding buffer (250 mM NaCl, 20 mM Tris-HCl, pH 7.4). The binding buffer containing BL21 cells was broken at high pressure, the broken product was centrifuged at 4°C and 9,000 $\times$ *g* for 20 min to remove the broken precipitate, and the supernatant was filtered with a 0.22-$\mu$m filter (Biosharp, China). ClyQ was purified by binding the 6$\times$His-tagged fusion protein supernatant to a His-Trap fast flow (FF) column (GE Healthcare Bio-Sciences AB, Uppsala, Sweden) and eluting with elution buffer (250 mM NaCl, 20 mM Tris-HCl, 200 mM imidazole, pH 7.4) according to the manufacturer's instructions. The purity and molecular weight of the protein were determined by sodium dodecyl sulfate-polyacrylamide gel electrophoresis (SDS-PAGE).

**Characterization of ClyQ.** The thermal stability of ClyQ was determined by incubating ClyQ at 37°C, 40°C, 42.5°C, and 45°C for 1 h. To determine the acid-base stability of ClyQ, different pH buffers (5.0, 6.0, 7.0, 8.0, 9.0, 10.0, and 11.0) were mixed with ClyQ at 37°C for 1 h. When incubating at different temperatures or pH environments, 100 $\mu$L of ClyQ (100 $\mu$g/mL) was taken out every 10 min and added into the same volume of *S. aureus* S3. After incubation at 37°C for 30 min, the OD$_{600}$ of the mixture was monitored. Meanwhile, the thermal stability of ClyQ (500 $\mu$g/mL) was monitored by nano-differential scanning fluorimetry (nanoDSF) using a Prometheus NT.48 instrument from NanoTemper Technologies (34). The metal ion stability of ClyQ was determined by incubating in PBS buffer (137 mM NaCl, 2.7 mM KCl, 4.3 mM Na$_2$HPO$_4$·H$_2$O, and 1.4 mM KH$_2$PO$_4$, pH 7.4) supplemented with 10, 100, and 1,000 $\mu$M Ca$^{2+}$, Mg$^{2+}$, Zn$^{2+}$, Ba$^{2+}$, and Mn$^{2+}$ for 1 h at 37°C; 100 $\mu$L of ClyQ (100 $\mu$g/mL) was then mixed with 100 $\mu$L of bacteria in 96-well plates. The plates were incubated at 37°C for 30 min, and the OD$_{600}$ was recorded.

**Bactericidal assays.** The host range of ClyQ and LysGH15 was determined by the turbidity reduction method (13, 35). Briefly, 100 $\mu$L of bacterial suspension and 100 $\mu$L of ClyQ or LysGH15 with a concentration of 100 $\mu$g/mL were mixed in a 96-well plate, and the OD$_{600}$ values of the mixtures were measured after incubation at 37°C for 30 min. For the dose-dependent assay, *S. aureus* ATCC 43300 and S3 were mixed with ClyQ at final concentrations ranging from 0.78 $\mu$g/mL to 50 $\mu$g/mL, and the mixture was incubated at 37°C for 1 h. The OD$_{600}$ was measured every 5 min, and PBS buffer was used to suspend bacteria and ClyQ.

To evaluate the antistaphylococcal activity of ClyQ and LysGH15, the number of bacterial colonies was calculated by mixing 100 $\mu$L of bacterial suspension with 100 $\mu$L of ClyQ and LysGH15 (100 $\mu$g/mL), and the mixture was incubated at 37°C for 1 h. Moreover, ClyQ was mixed with *S. aureus* ATCC 43300 and S3 and incubated at 37°C for 60 min, and bacterial cells were monitored at 20-min intervals. Bacterial cell numbers were calculated by plate counting of a 10-fold serial dilution.

**MIC of ClyQ.** The broth dilution method was used to determine the MIC of ClyQ (36). Bacteria were diluted in 0.9% normal saline to a standard of 0.5 McIntosh turbidity and further diluted to 1 $\times$ 10$^5$ CFU/ mL by the addition of cation-adjusted Mueller-Hinton broth (CAMHB; Becton, Dickinson, Franklin Lakes,

NJ) (37). Then, 100 $\mu$L of bacteria suspension was added with 100 $\mu$L of ClyQ at different concentrations (1 to 512 mg/L) in 96-well plates. The results were observed after incubation at 37°C for 18 h. The MIC was defined as the lowest protein concentration inhibiting visible bacteria growth. The experiment was repeated three times.

**Antibiofilm activity of ClyQ.** The 96-well plates were used to test the antibiofilm ability of ClyQ, as previously described (33, 38). The bacteria concentration was adjusted to $1 \times 10^6$ CFU/mL, and 200 $\mu$L of the bacteria solution was added to 96-well plates and incubated for 24 or 48 h at 30°C. Subsequently, the biofilms were treated with 200 $\mu$L of ClyQ (0, 5, and 40 $\mu$g/mL) at 37°C for 1 h. Finally, the biomass of biofilms was determined by crystal violet (1.0% [wt/vol]) staining, and bacterial burden in biofilms was determined by plate counting of a 10-fold serial dilution.

**SEM images of S. aureus biofilms.** The biofilms of *S. aureus* S3 and ATCC 43300 were grown in 500 $\mu$L of LB medium on 24-well plates with coverslips at 30°C for 48 h. The 24-well plates were then treated with ClyQ at a concentration of 40 $\mu$g/mL at 37°C for 1 h, and PBS was used as a negative control. After the treatment, plates were washed twice with PBS, fixed with 2% glutaraldehyde for 2 h, and washed twice with PBS. Finally, the coverslips were sprayed with gold, and images of *S. aureus* were acquired by FESEM (magnifications of ×2,000 and ×5,000; NTC, Japan).

**Ethics statement.** All animal experiments were approved by the Laboratory Animal Monitoring Committee of Huazhong Agricultural University and were performed according to the corresponding guidelines for laboratory animal operations in Huazhong Agricultural University. The corresponding ethical approval code is HZAUMO-2022-0057.

**Mouse infection model.** The systemic infection models were performed as described previously (16, 39). Specific pathogen-free (SPF) BALB/c female mice (6 weeks old) were purchased from the Experimental Animal Centre of Huazhong Agricultural University, China. First, different concentrations of *S. aureus* ATCC 29213 ($1 \times 10^8$, $5 \times 10^8$, $8 \times 10^8$, and $1 \times 10^9$ CFU/mouse) were injected intraperitoneally to determine the $MLD_{100}$ of ATCC 29213. In the mouse systemic infection model, ATCC 29213 was injected intraperitoneally ($8 \times 10^8$ CFU/mouse; $1 \times MLD_{100}$), and organs (heart, liver, spleen, lung, and kidney) were weighed and homogenized in 1 mL of PBS for 5 min at 60 reps (Retsch MM400) at 1 h and 3 h after the challenge. Blood and homogenates were serially diluted and plated on LB agar for the assessment of bacterial abundance. Plates were incubated at 37°C overnight, and numbers of visible bacteria were counted. For survival rate experiments, each group ($n = 5$) of mice was intraperitoneally injected with different concentrations of ClyQ (200 and 500 $\mu$g/mouse), 15 mg/kg vancomycin, or PBS (200 $\mu$L) at 1 h postinfection, and mice were intraperitoneally injected with various concentrations of ClyQ (500 $\mu$g/mouse and 1 mg/mouse), 15 mg/kg vancomycin, or PBS (200 $\mu$L) at 3 h postinfection. Uninfected mice were intraperitoneally injected with 1 mg/mouse ClyQ ($n = 5$) and 15 mg/kg vancomycin ($n = 5$) as the negative control. Mouse survival rates were observed for 7 days.

**Mouse skin infection model.** A mouse model of skin infection was performed as described previously, with minor modifications (40). An area of skin (approximately 2 cm$^2$) on the dorsal side of each mouse was stripped using 3M autoclaved tapes 20 times. Mouse skin was treated with PBS as a vehicle control and with 1 $\mu$g/mouse mupirocin (2%, 50 $\mu$L) as a positive control (12). The concentration of *S. aureus* S3 was controlled to $5 \times 10^8$ CFU/mL, and 20 $\mu$L ($1 \times 10^7$ CFU) of the bacterial solution was added to the wound area. Subsequently, the mice were divided into four groups ($n = 9$) and treated with PBS, ClyQ (50 $\mu$g/mouse), mupirocin (1 $\mu$g/mouse), and the combination of ClyQ (50 $\mu$g/mouse) and mupirocin (1 $\mu$g/mouse), respectively, at 20, 23, 26, 36, 39, and 42 h after infection. Mice were euthanized at 43 h postinfection, and the injured skin was homogenized. Homogenates were serially diluted and plated on LB agar plates for visible cell counting. Simultaneously, the skin samples were fixed with a 4% paraformaldehyde solution for histopathological assessment. Concentrations of TNF-$\alpha$ and IL-6 in the skin tissue were determined using Quantikine mouse TNF-$\alpha$/IL-6 commercial kits (NeoBioscience, Inc., Shenzhen, China).

**Checkerboard assay.** The checkerboard assay for the combination of ClyQ and mupirocin was performed in 96-well plates (6). Concentration gradients between ClyQ and mupirocin were prepared in the horizontal and vertical directions, and PBS was used as a negative control. The method of preparing bacteria was the same as for MIC determination, as described above, and the bacteria were added to each well. The plates were incubated at 37°C for 18 h. Subsequently, the MIC was recorded. The values of the fractional inhibitory concentration index (FICI) were calculated according to the previous literature (30).

**Resistance development assay.** The development of bacterial resistance to ClyQ and mupirocin was tested according to previous methods, with minor modifications (12, 41, 42). *S. aureus* S3 and ATCC 29213 were exposed to increasing concentrations (1/32 × MIC to 4 × MIC) of ClyQ, mupirocin, and a combination of ClyQ and mupirocin. Each culture generation was divided into two aliquots. One aliquot received CAMHB with the next concentration of ClyQ, mupirocin, and a combination of ClyQ and mupirocin and was incubated at 37°C for 12 h. The other aliquot was plated on MHB agar plates that were treated with 1 × MIC of ClyQ, mupirocin, and a combination of ClyQ and mupirocin. Three colonies grown on plates were used to determine the MIC values of ClyQ, mupirocin, and a combination of ClyQ and mupirocin. The process was repeated for 8 rounds.

**Statistical analysis.** The experimental data are presented as mean ± standard deviation (SD). Statistical significance was determined using an unpaired Student's *t* test. A *P* value of <0.05 was considered a statistically significant difference. GraphPad Prism (version 9) was used for all statistical analyses.

## SUPPLEMENTAL MATERIAL

Supplemental material is available online only.

**SUPPLEMENTAL FILE 1**, PDF file, 0.4 MB.

## ACKNOWLEDGMENTS

This work was supported by grants from the National Program on Key Research Project of China (2021YFD1800300 and 2018YFD0500204), "Yingzi Tech & Huazhong Agricultural University Intelligent Research Institute of Food Health" (numbers IRIFH202209 and IRIFH202302), and the Fundamental Research Funds for the Central Universities (IRIFH202301).

X.-c.D. and X.-x.L. drafted the main manuscript and performed the data analysis. X.-c.D., X.-x.L., S.W., and F.-q.Z. planned and performed experiments. X.-m.L. and P.Q. were responsible for the experimental design. All authors reviewed and agreed on publication of the manuscript.

We declare no conflicts of interest.

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
