## [Reviewer comments · Microbiology Spectrum]

Microbiology Spectrum

Exploiting broad-spectrum chimeric lysin to cooperate with mupirocin against *Staphylococcus aureus*-induced skin infections and delay the development of mupirocin resistance

Duan Xiaochao, Li Xinxin, Li Xiangmin, Wang Shuang, Zhang Fenqiang, and Ping Qian

Corresponding Author(s): Ping Qian, Huazhong Agricultural University

Review Timeline:

Submission Date:	December 8, 2022
Editorial Decision:	February 15, 2023
Revision Received:	March 30, 2023
Accepted:	April 10, 2023

Editor: Cezar Khursigara

Reviewer(s): The reviewers have opted to remain anonymous.

Transaction Report:

DOI: <https://doi.org/10.1128/spectrum.05050-22>

February 15, 2023

Dr. Qian Ping
State Key Laboratory of Agricultural Microbiology
Hongshan District Shizishan Street 1#
wuhan 430070
China

Re: Spectrum05050-22 (Exploiting broad-spectrum chimeric lysin to cooperate with mupirocin against *Staphylococcus aureus*-induced skin infections and delay the development of mupirocin resistance)

Dear Dr. Qian Ping:

Two experts have reviewed your manuscript, and although they agree that it has merit, modifications are required before it can be considered for publication. Please address all the comments when submitting a revised manuscript.

Thank you for submitting your manuscript to *Microbiology Spectrum*. When submitting the revised version of your paper, please provide (1) point-by-point responses to the issues raised by the reviewers as file type "Response to Reviewers," not in your cover letter, and (2) a PDF file that indicates the changes from the original submission (by highlighting or underlining the changes) as file type "Marked Up Manuscript - For Review Only". Please use this link to submit your revised manuscript - we strongly recommend that you submit your paper within the next 60 days or reach out to me. Detailed instructions on submitting your revised paper are below.

Link Not Available

Below you will find instructions from the *Microbiology Spectrum* editorial office and comments generated during the review.

Sincerely,

Cezar Khursigara

Editor, *Microbiology Spectrum*

Journals Department
Reviewer comments:

Reviewer #1 (Comments for the Author):

The manuscript by Duan et al. concerns the antibacterial effect of a novel hybrid lysin composed of catalytic and cell wall-binding domain of *Staphylococcus* phage lysin and lysin of *Enterococcus faecalis* phage, and constructed by the authors. The authors showed a broad lytic range of their hybrid endolysin against living cells of staphylococci of various species and also against *E. faecalis*, *Erysipelothrix rhusiopathiae* and various species of streptococci under in vitro conditions. They tested the efficacy of this lysin in rescuing mice infected with a lethal dose of *S. aureus* and in decreasing the living *S. aureus* cell counts on a skin of infected mice. The most interesting results concern the observation of synergistic effect of the hybrid lysin and mupirocin. A

panel of methods used by the authors is not novel and mostly resembles that described earlier by Yang et al. 2017 (doi: 10.1038/srep17257) to characterize one of the chimeric lytic enzyme active against most of the streptococcal strains tested, but also against representatives of staphylococcal and enterococcal species. Additionally, the authors demonstrated that the resistance development to their hybrid lysin and to mupirocin occurs later when both these compounds are used together, as compared to their separate use. The discussion section contains mostly repetition of conclusions from the results section. Comparison of the structure and activities of the hybrid lysin described by the authors with other hybrid lysins active against staphylococci and/or streptococci/enterococci should be included in the discussion, including the synergies of antibacterial effects of other lysins with antibiotics. Overall the manuscript describes a hybrid lysin of potent antistaphylococcal properties and is of potential interest for the readers of Microbiology Spectrum, but contains certain faults, including spelling errors and poor English.

The Materials and Methods section should allow to reproduce the described experiments. Meanwhile, the methods descriptions provided by the authors are in many cases too much abbreviated. They should be supplemented with missing data (see e.g., L. 285, 298-299, 309, 311, 341-345).

L. 53 and elsewhere in the manuscript: Please italicize genus and species names.

L. 117. What do you mean by "visible bacterial number" Do you mean colonies grown upon plating the ClyQ treated bacterial suspension? This should be specified.

L. 273: Replace lysGH15 with LysGH15

L. 309: Replace "solution" with "suspension". What buffer was used to suspend bacteria and ClyQ.

L. 325: What do you mean by "cation adjusted"

L. 329: Was it estimated simply by eye? If so the reliability and reproducibility of results is questionable.

L. 601: How bacterial abundance in various mouse organs was assayed. This should be specified.

Figure 3E is not convincing. The authors should provide more than just one biofilm image following the ClyQ treatment of bacteria, at least in supplementary data. How do the authors know that e.g., bottom right panel of Figure 3E does not show an artifact?

Figure 4 and the legend to this Figure need redrawing and rewriting, respectively. Explanations of symbols to the left of graphs in Figure 4A,C and D only partly correspond to the graphs shown. E.g., in Figure 4A the dotted line representing 1×10^9 S. aureus cells per mouse is missing. Without correcting colors some dotted lines in Figure C and D are difficult to differentiate. The legend to Figure 4 is unclear (e.g., "the survival rate of mice was intraperitoneally treated with vancomycin"). What was the effect of ClyQ alone on the viability of mice. Was it tested?

Figure 5. The bottom left panel of Figure 5 shows the skin section in reverse orientation as compared to other panels> this should be corrected for better clarity of this Figure. Additionally the authors should include in this figure the results of relevant measurements (cfu, TNF- α and IL-6) performed with the use of uninfected skin. Without such controls it is unclear how far the values of measurements return to normal upon treatment of infected skin with ClyQ or mupirocin or both.

The manuscript should be corrected by a native English speaker.

Reviewer #2 (Comments for the Author):

This study describes the combination effect of chimeric lysin and mupirocin on the inhibition of Staphylococcus aureus planktonic and biofilm cells in vitro and in vivo. There are some doubts that are needed to be stated.

1. Since chimeric lysin has been constructed to broaden the host range and antimicrobial activity, these host range and antimicrobial activity of ClyO should be compared with phage lysin.

2. State any criteria for using mupirocin in this study.

3. In Fig. 6, "0" concentrations of ClyO and mupirocin should be added on checkerboard.

4. Further discussion is needed on the results that show more ClyO mutant development in the combination than single treatment (Fig. 7D)

Staff Comments:

Preparing Revision Guidelines

To submit your modified manuscript, log onto the eJP submission site at <https://spectrum.msubmit.net/cgi-bin/main.plex>. Go to Author Tasks and click the appropriate manuscript title to begin the revision process. The information that you entered when you

first submitted the paper will be displayed. Please update the information as necessary. Here are a few examples of required updates that authors must address:

Please return the manuscript within 60 days; if you cannot complete the modification within this time period, please contact me. If you do not wish to modify the manuscript and prefer to submit it to another journal, please notify me of your decision immediately so that the manuscript may be formally withdrawn from consideration by Microbiology Spectrum.

Responses to the reviewers' comments

Dear editor,

On behalf of the co-authors, I would like to express our great appreciation to you and the reviewers. Thank you very much for your feedback regarding our manuscript entitled "Exploiting broad-spectrum chimeric lysin to cooperate with mupirocin against *Staphylococcus aureus*-induced skin infections and delay the development of mupirocin resistance" (Manuscript ID: Spectrum05050-22). We have carefully revised our paper according to the reviewers' comments and have responded to all questions point-by-point. Based on the instructions provided in your letter, we have uploaded the file of the revised manuscript. We hope that these corrections will be met with approval.

Responds to the reviewers' comments:

Reviewer comments:

Reviewer #1:

The manuscript by Duan et al. concerns the antibacterial effect of a novel hybrid lysin composed of catalytic and cell wall-binding domain of *Staphylococcus* phage lysin and lysin of *Enterococcus faecalis* phage, and constructed by the authors. The authors showed a broad lytic range of their hybrid endolysin against living cells of staphylococci of various species and also against *E. faecalis*, *Erysipelothrix rhusiopathiae* and various species of streptococci under in vitro conditions. They tested the efficacy of this lysin in rescuing mice infected with a lethal dose of *S. aureus* and in decreasing the living *S. aureus* cell counts on a skin of infected mice. The most interesting results concern the observation of synergistic effect of the hybrid lysin and mupirocin. A panel of methods used by the authors is not novel and mostly resembles that described earlier by Yang et al. 2017 (doi: 10.1038/srep17257)

to characterize one of the chimeric lytic enzyme active against most of the streptococcal strains tested, but also against representatives of staphylococcal and enterococcal species. Additionally, the authors demonstrated that the resistance development to their hybrid lysin and to mupirocin occurs later when both these compounds are used together, as compared to their separate use.

The discussion section contains mostly repetition of conclusions from the results section. Comparison of the structure and activities of the hybrid lysin described by the authors with other hybrid lysins active against staphylococci and/or streptococci/enterococci should be included in the discussion, including the synergies of antibacterial effects of other lysins with antibiotics.

Reply: Thank you very much for your reading and comments of our manuscript. The repetition of conclusions from the results section have been condensed as suggested.

We have added the comparison of the structure and activities of ClyQ with other chimeric lysin. Please see lines 239-251. Meanwhile, we have discussed antibacterial effects of combination of other lysins and antibiotics in the discussion. Please see line 252-266.

Overall the manuscript describes a hybrid lysin of potent antistaphylococcal properties and is of potential interest for the readers of Microbiology Spectrum, but contains certain faults, including spelling errors and poor English.

Reply: Thank you for your suggestion. We apologize for the poor language of our manuscript. We have checked the entire manuscript carefully and corrected this issue as suggested. We have also involved native English speakers for language corrections.

The Materials and Methods section should allow to reproduce the described experiments. Meanwhile, the methods descriptions provided by the authors are in many cases too much abbreviated. They should be supplemented with

missing data (see e.g., L. 285, 298-299, 309, 311, 341-345).

Reply: Thank you for your suggestion. We have described these methods as suggested in the revised manuscript in detail. Please see line 330, 343-346, 357, 360, 391-396.

L. 53 and elsewhere in the manuscript: Please italicize genus and species names.

Reply: Thank you for your suggestions. We have corrected it as suggested in the revised manuscript.

L. 117. What do you mean by "visible bacterial number" Do you mean colonies grown upon plating the ClyQ treated bacterial suspension? This should be specified.

Reply: We are grateful to you for pointing out this problem. As the reviewer mentioned, to evaluate the reduction of viable cell numbers by ClyQ against different staphylococci strains, the number of bacterial colonies was calculated by mixing 100 μ L of bacterial suspension with 100 μ L of ClyQ (100 μ g/mL), and the mixture was incubated at 37°C for 1 h. Then, bacterial cells were calculated by plate counting of 10-fold serial dilution. We have corrected it as suggested in the revised manuscript. Please see line 148.

L. 273: Replace lysGH15 with LysGH15

Reply: We have corrected it in the revised manuscript.

L. 309: Replace "solution" with "suspension". What buffer was used to suspend bacteria and ClyQ.

Reply: Thank you for your suggestion. All "solution" have been changed to "suspension". Meanwhile, phosphate buffer saline (PBS) (137 mM NaCl, 2.7 mM KCl, 4.3 mM Na₂HPO₄·H₂O, 1.4 mM KH₂PO₄, pH 7.4) buffer was used to suspend bacteria and ClyQ. We have corrected it in the revised manuscript.

L. 325: What do you mean by "cation adjusted"

Reply: Thank you for your suggestion. The "Cation-Adjusted Mueller-Hinton broth (CAMHB)" implies adjusting the amount of Ca^{2+} and Mg^{2+} ions in Mueller-Hinton broth (MHB) (1). Because the Clinical and Laboratory Standards Institute (CLSI) 2018 (2) recommends the determination of MIC of staphylococci by CAMHB, we chose the commercial CAMHB (Becton Dickinson, Franklin Lakes, NJ) to test the MIC. We have added the source of CAMHB and reference in the revised manuscript.

L. 329: Was it estimated simply by eye? If so the reliability and reproducibility of results is questionable.

Reply: We are grateful to you for pointing out this issue. The MIC is defined as the lowest concentration of the antimicrobial agent that inhibits visible growth of the tested isolate as observed with the unaided eye (1), and previous studies have shown that this method can be used to determine the MIC of phage lysin (3, 4). According to our results and previous research (5), no visible growth of the tested isolate was observed at the bottom of 96-well plate under the minimum inhibitory concentration, but when the concentration of lysin is greater than MIC, there is obvious bacterial growth. In addition, the results of experiment repeated three times are consistent. We have added the references that have used the method.

L. 601: How bacterial abundance in various mouse organs was assayed. This should be specified.

Reply: Thank you for your suggestion. In the mouse systemic infection model, ATCC29213 was injected intraperitoneally with 8×10^8 cfu/mouse ($1 \times \text{MLD}_{100}$), and organs (heart, liver, spleen, lung, and kidney) were weighed and homogenized in 1 ml PBS for 5 min at 60 reps (Retsch MM400) at 1 h and 3 h after the challenge. Blood and homogenates were serially diluted and plated

on LB Agar for bacterial abundances. The plates were incubated at 37°C overnight and the number of visible bacteria were counted. We have corrected it as suggested in the method section of "Mouse infection model".

Figure 3E is not convincing. The authors should provide more than just one biofilm image following the ClyQ treatment of bacteria, at least in supplementary data. How do the authors know that e.g., bottom right panel of Figure 3E does not show an artifact?

Reply: We thank the reviewer for pointing out this important issue. We have repeated the experiment of Figure 3E by Field Emission Scanning Electron Microscope (FESEM). In order to show the complete biofilm, our results contains 2,000× and 5,000× images of biofilms. The part of results was placed in the new figure 4. Please see Figure 4. We have also rewritten the method section of "SEM images of *S. aureus* biofilms".

Figure 4 and the legend to this Figure need redrawing and rewriting, respectively. Explanations of symbols to the left of graphs in Figure 4A,C and D only partly correspond to the graphs shown. E.g., in Figure 4A the dotted line representing 1×10^9 *S. aureus* cells per mouse is missing. Without correcting colors some dotted lines in Figure C and D are difficult to differentiate.

Reply: Thank you for your valuable suggestion. The figure 4a, c, and d have been redrawn as suggested. Please see Figure 5.

The legend to Figure 4 is unclear (e.g., "the survival rate of mice was intraperitoneally treated with vancomycin"). What was the effect of ClyQ alone on the viability of mice. Was it tested?

Reply: Thank you for pointing out this issue. Figure 4C and D have shown that effect of ClyQ and vancomycin on survival rate of mice infected with $1 \times \text{MLD}_{100}$ *S. aureus* ATCC29213. Mice were intraperitoneally treated with PBS,

vancomycin (15 mg/kg), and different concentrations of ClyQ at 1h (Fig 5C) and 3h (Fig 5D) post-infection. We have corrected the figure legend in the revision. In revised manuscript, we have tested the effect of ClyQ and vancomycin alone on the viability of the uninfected mice, and the survival rate of the uninfected mice intraperitoneally injected with ClyQ and vancomycin alone was recorded for 7 days (Fig 5D). The results indicated that ClyQ (1mg/mouse) and vancomycin (15mg/kg) could not cause the death of mice.

Figure 5. The bottom left panel of Figure 5 shows the skin section in reverse orientation as compared to other panels> this should be corrected for better clarity of this Figure.

Reply: Thank you for your suggestion. We have corrected it as suggested in revision. Please see Figure 6D.

Additionally the authors should include in this figure the results of relevant measurements (cfu, TNF- α and IL-6) performed with the use of uninfected skin. Without such controls it is unclear how far the values of measurements return to normal upon treatment of infected skin with ClyQ or mupirocin or both.

Reply: Thank you for your useful suggestion. We have supplied related experiments. Concentrations of TNF- α and IL-6 reached 56.8 and 92.9 pg/mL in uninfected mouse skin, respectively. The results have added into Figure 6B and 6C. Please see Figure 6B and 6C.

The manuscript should be corrected by a native English speaker.

Reply: Thank you for your suggestion. We have checked the entire manuscript carefully and corrected this issue as suggested. We have also involved native English speakers for language corrections.

Reviewer #2 (Comments for the Author):

This study describes the combination effect of chimeric lysin and mupirocin on the inhibition of *Staphylococcus aureus* planktonic and biofilm cells in vitro and in vivo. There are some doubts that are needed to be stated.

Reply: Thank you very much for your reading and comments of our manuscript.

1. Since chimeric lysin has been constructed to broaden the host range and antimicrobial activity, these host range and antimicrobial activity of ClyO should be compared with phage lysin.

Reply: We want to thank reviewer for constructive advice. We have supplied related experiments as suggested. In this study, we want to explore whether the combination of chimeric lysin and mupirocin can better treat *S. aureus* infection and delay the development of mupirocin resistance compared to chimeric lysin and mupirocin alone. In order to obtain broad-spectrum phage lysin which possess excellent antistaphylococcal activity, we constructed the chimeric lysin ClyQ. Previous research indicated that *S. aureus* phage lysin LysGH15 possess exceptionally lytic activity against *S. aureus* (6), and ClyQ share the same CHAP domain with LysGH15. Therefore, the host range and antimicrobial activity of ClyQ was compared with LysGH15. Our results have shown that ClyQ and LysGH15 can lyse the staphylococci. In contrast to ClyQ, LysGH15 could not lyse the streptococci, *Enterococcus faecalis*, and *Erysipelothrix rhusiopathiae*. Our results demonstrated that ClyQ possess a broad host range and exceptionally lytic activity. Please see figure 1.

2. State any criteria for using mupirocin in this study.

Reply: Thank you for your suggestion. At present, *S. aureus*-induced skin infection can be treated by commercially available mupirocin (2%) (7). Therefore, we have used the 2% (w/v) mupirocin as a positive group and compared the therapeutic effect with ClyQ. In the study, 1 µg/mouse of

mupirocin (2%, 50 μ l) was used to treat skin infection. We have added the related reference in revised manuscript. Please see line 429-430.

3. In Fig. 6, "0" concentrations of ClyO and mupirocin should be added on checkerboard.

Reply: Thank you for your suggestion. Based on our results and related references (8), the values of FICI were calculated by the MIC of single ClyQ or mupirocin and the combination ClyQ and mupirocin. The group of "0 concentrations of ClyO and mupirocin" can be negative group in this experiment, and the group cannot be calculated as FICI. Thus, the group was not shown on the figure.

4. Further discussion is needed on the results that show more ClyO mutant development in the combination than single treatment (Fig. 7D)

Reply: We want to thank the reviewer for this constructive advice. We have added the related discussion as suggested in the revised manuscript. Please see line 266-272.

Editor:

Two experts have reviewed your manuscript, and although they agree that it has merit, modifications are required before it can be considered for publication. Please address all the comments when submitting a revised manuscript.

Reply: Thank you very much for your feedback. We have carefully revised our paper according to the reviewers' comments and have responded to all questions point-by-point. Based on the instructions provided in your letter, we have uploaded the file of the revised manuscript. We hope that these corrections will be met with approval.

References

1. Wiegand I, Hilpert K, Hancock RE. 2008. Agar and broth dilution methods to determine the minimal inhibitory concentration (MIC) of antimicrobial substances. *Nat Protoc* 3:163-75.
2. Institute CaLS. 2018. Methods for dilution antimicrobial susceptibility tests for bacteria that grow aerobically—11th ed. Clinical and Laboratory Standards Institute, Wayne, PA.,
3. Lood R, Raz A, Molina H, Euler CW, Fischetti VA. 2014. A highly active and negatively charged *Streptococcus pyogenes* lysin with a rare D-alanyl-L-alanine endopeptidase activity protects mice against streptococcal bacteremia. *Antimicrob Agents Chemother* 58:3073-84.
4. Shan Y, Yang N, Teng D, Wang X, Mao R, Hao Y, Ma X, Fan H, Wang J. 2020. Recombinant of the Staphylococcal Bacteriophage Lysin CHAP(k) and Its Elimination against *Streptococcus agalactiae* Biofilms. *Microorganisms* 8.
5. Li XX, Zhang FQ, Wang S, Duan XC, Hu DY, Gao DY, Tao P, Li XM, Qian P. 2023. *Streptococcus suis* prophage lysin as a new strategy for combating streptococci-induced mastitis and *Streptococcus suis* infection. *J Antimicrob Chemother* 78:747-756.
6. Gu J, Xu W, Lei L, Huang J, Feng X, Sun C, Du C, Zuo J, Li Y, Du T, Li L, Han W. 2011. LysGH15, a novel bacteriophage lysin, protects a murine bacteremia model efficiently against lethal methicillin-resistant *Staphylococcus aureus* infection. *J Clin*

Microbiol 49:111-7.

7. Pastagia M, Euler C, Chahales P, Fuentes-Duculan J, Krueger JG, Fischetti VA. 2011. A novel chimeric lysin shows superiority to mupirocin for skin decolonization of methicillin-resistant and -sensitive *Staphylococcus aureus* strains. *Antimicrob Agents Chemother* 55:738-44.
8. Shao Z, Wulandari E, Lin RCY, Xu J, Liang K, Wong EHH. 2022. Two plus One: Combination Therapy Tri-systems Involving Two Membrane-Disrupting Antimicrobial Macromolecules and Antibiotics. *ACS Infect Dis* 8:1480-1490.

April 10, 2023

Prof. Ping Qian
Huazhong Agricultural University
Wuhan
China

Re: Spectrum05050-22R1 (Exploiting broad-spectrum chimeric lysin to cooperate with mupirocin against Staphylococcus aureus-induced skin infections and delay the development of mupirocin resistance)

Dear Prof. Ping Qian:

Your manuscript has been accepted, and I am forwarding it to the ASM Journals Department for publication. You will be notified when your proofs are ready to be viewed.

Sincerely,

Cezar Khursigara
Editor, Microbiology Spectrum
